# Real-time decoding of question-and-answer speech dialogue using human cortical activity

David A. Moses[1], Matthew K. Leonard[1], Joseph G. Makin[1] & Edward F. Chang[1]

Natural communication often occurs in dialogue, differentially engaging auditory and sensorimotor brain regions during listening and speaking. However, previous attempts to decode speech directly from the human brain typically consider listening or speaking tasks in isolation. Here, human participants listened to questions and responded aloud with answers while we used high-density electrocorticography (ECoG) recordings to detect when they heard or said an utterance and to then decode the utterance's identity. Because certain answers were only plausible responses to certain questions, we could dynamically update the prior probabilities of each answer using the decoded question likelihoods as context. We decode produced and perceived utterances with accuracy rates as high as 61% and 76%, respectively (chance is 7% and 20%). Contextual integration of decoded question likelihoods significantly improves answer decoding. These results demonstrate real-time decoding of speech in an interactive, conversational setting, which has important implications for patients who are unable to communicate.

[1] Department of Neurological Surgery and the Center for Integrative Neuroscience at UC San Francisco, 675 Nelson Rising Lane, San Francisco, CA 94158, USA. Correspondence and requests for materials should be addressed to E.F.C. (email: Edward.Chang@ucsf.edu)

The lateral surface of the human cortex contains neural populations that encode key representations of both perceived and produced speech[1–9]. Recent investigations of the underlying mechanisms of these speech representations have shown that acoustic[10–12] and phonemic[4,13,14] speech content can be decoded directly from neural activity in superior temporal gyrus (STG) and surrounding secondary auditory regions during listening. Similarly, activity in ventral sensorimotor cortex (vSMC) can be used to decode characteristics of produced speech[5–9,15–17], based primarily on kinematic representations of the supralaryngeal articulators[7–9] and the larynx for voicing and pitch[17]. A major challenge for these approaches is achieving high single-trial accuracy rates, which is essential for a clinically relevant implementation to aid individuals who are unable to communicate due to injury or neurodegenerative disorders.

Recently, speech decoding paradigms have been implemented in real-time applications, including the ability to map speech-evoked sensorimotor activations[18], generate neural encoding models of perceived phonemes[19], decode produced isolated phonemes[20], detect voice activity[21], and classify perceived sentences[14]. These demonstrations are important steps toward the development of a functional neuroprosthesis for communication that decodes speech directly from recorded neural signals. However, to the best of our knowledge there have not been attempts to decode both perceived and produced speech from human participants in a real-time setting that resembles natural communication. Multimodal decoding of natural speech may have important practical implications for individuals who are unable to communicate due to stroke, neurodegenerative disease, or other causes[22,23]. Despite advances in the development of assistive communication interfaces that restore some communicative capabilities to impaired patients via non-invasive scalp electroencephalography[24], invasive microelectrode recordings[25], electrocorticography (ECoG)[26], and eye tracking methodologies[27], to date there is no speech prosthetic system that allows users to have interactions on the rapid timescale of human conversation.

Here we demonstrate real-time decoding of perceived and produced speech from high-density ECoG activity in humans during a task that mimics natural question-and-answer dialogue (see Supplementary Movie 1). While this task still provides explicit external cueing and timing to participants, the interactive and goal-oriented aspects of a question-and-answer paradigm represent a major step towards more naturalistic applications. During ECoG recording, participants first listened to a set of pre-recorded questions and then verbally produced a set of answer responses. These data served as input to train speech detection and decoding models. After training, participants performed a task in which, during each trial, they listened to a question and responded aloud with an answer of their choice. Using only neural signals, we detect when participants are listening or speaking and predict the identity of each detected utterance using phone-level Viterbi decoding. Because certain answers are valid responses only to certain questions, we integrate the question and answer predictions by dynamically updating the prior probabilities of each answer using the preceding predicted question likelihoods. Incorporating both modalities significantly improves answer decoding performance. These results demonstrate reliable decoding of both perceived and produced utterances in real-time, illustrating the promise of neuroprosthetic speech systems for individuals who are unable to communicate.

## Results

### Overview of the real-time decoding approach.
While participants performed a question-and-answer natural speech perception (Fig. 1a) and production (Fig. 1b) task, we acquired neural activity from high-density ECoG arrays that covered auditory and sensorimotor cortical regions. In real-time, neural activity was filtered to extract signals in the high gamma frequency range (70–150 Hz; Fig. 1c, Supplementary Fig. 1), which correlate with multi-unit activity[28] and have been previously used to decode speech signals from auditory[4,10,13,14] and sensorimotor[5,7,8,15,16] brain regions. We used these high gamma signals to perform real-time speech event detection, predicting which time segments of the neural activity occurred during question perception (Fig. 1d, blue curve) or answer production (Fig. 1d, red curve). The speech event detector was trained to identify spatiotemporal neural patterns associated with these events, such as rapid evoked responses in STG during question perception or causal activity patterns in vSMC during answer production, which were used during real-time decoding to predict the temporal onsets and offsets of detected speech events (see Supplementary Fig. 2 and Section 4.6.1 for more details on the event detection procedure).

For each time segment that was labeled as a question event, a classification model was used to analyze the high gamma activity and compute question likelihoods using phone-level Viterbi decoding[29] (Fig. 1e). In this approach, a hidden Markov model (HMM) was used to represent each question utterance and estimate the probability of observing a time segment of high gamma activity assuming that the participant was hearing the sequence of phones that comprise the utterance. The most likely question was output as the decoded question (Fig. 1f).

We hypothesized that answer decoding could be improved by utilizing knowledge about the previously decoded question. We designed this question-and-answer task such that specific answer responses were only valid for certain questions (Table 1). For example, if a participant heard the question "How is your room currently?", there were five valid answers ("Bright", "Dark", "Hot", "Cold", and "Fine"). We used the relationship between each question and the valid answers to define context priors (Fig. 1g), which were represented by a flat probability distribution for within-question answers and zero probability for out-of-question answers. A context integration model combined these context priors with decoded question likelihoods to compute answer prior probabilities (Fig. 1h). This context integration model was used during online real-time decoding and offline analysis (except where specifically indicated).

As with question decoding, for each time segment that was labeled as an answer event, a classification model was used to analyze the high gamma activity and compute answer likelihoods using phone-level Viterbi decoding (Fig. 1i). The context integration model combined these answer likelihoods with the answer priors to obtain answer posterior probabilities (Fig. 1j), and the answer with the highest posterior probability was output as the decoded answer (Fig. 1k).

Prior to testing, models were fit using data collected during separate training task blocks. The question classification models were fit using data collected while participants listened to multiple repetitions of each of the question stimuli, and the answer classification models were fit using data collected while participants read each answer aloud multiple times. The speech detection models were fit using both of these types of training task blocks. Information about the amount of data collected for training and testing with each participant is provided in Supplementary Table 1.

### Question and answer decoding performance.
In offline analysis using the real-time decoding approach, we evaluated decoding accuracy for questions, answers without context integration, and

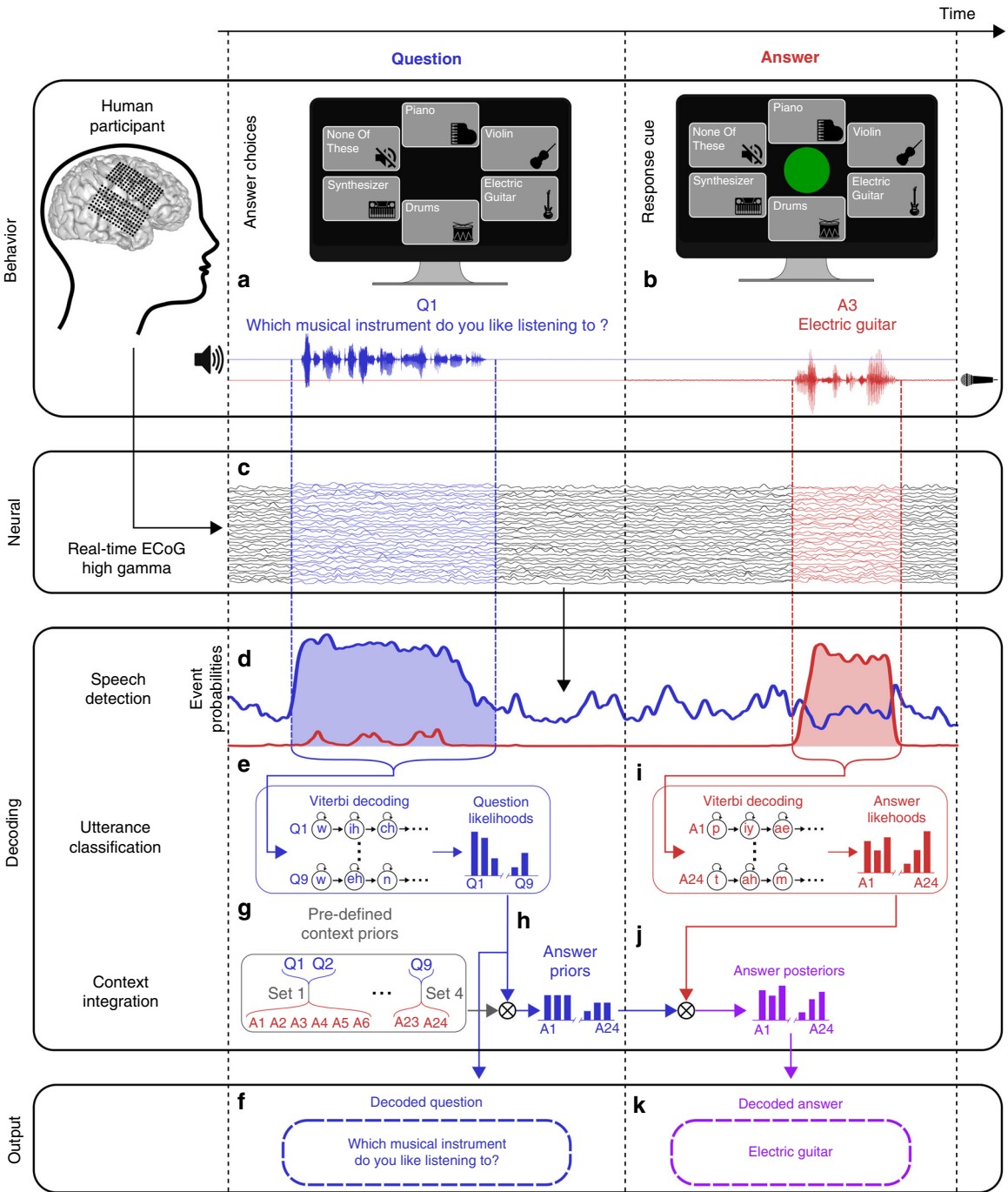

**Fig. 1** Schematic of real-time speech decoding during a question (blue) and answer (red) task. **a** On each trial, participants hear a question and see a set of possible answer choices on a screen. **b** Participants are instructed to freely choose and verbally produce one of the answers when a green response cue appears on the screen. **c** Simultaneously, cortical activity is acquired from ECoG electrodes implanted across temporal and frontal cortex and then filtered in real-time to extract high gamma activity. **d** A speech detection model uses the spatiotemporal pattern of high gamma activity to predict whether a question is being heard or an answer is being produced (or neither) at each time point. **e** When the speech detection model detects a question event, that time window of high gamma activity is passed to a question classifier that uses phone-level Viterbi decoding to compute question utterance likelihoods. **f** The question with the highest likelihood is output as the decoded question. **g** To integrate questions and answers, the stimulus set was designed such that each answer was only likely for certain questions (context priors). **h** These context priors are combined with the predicted question likelihoods to obtain answer priors. **i** When the speech detection model detects an answer event, that time window of neural activity is passed to an answer classifier that uses phone-level Viterbi decoding to compute answer utterance likelihoods. **j** The context integration model combines these answer likelihoods with the answer priors to yield answer posterior probabilities (purple). **k** The answer with the highest posterior probability is output as the decoded answer. The answer choice icons shown in (**a**, **b**) were made by www.freepik.com from www.flaticon.com

**Table 1 The question/answer sets**

| QA set number | Question | Answer |
|---|---|---|
| 1 | Which musical instrument do you like listening to? | Piano |
|  | Which musical instrument do you dislike hearing? | Violin |
|  |  | Electric guitar |
|  |  | Drums |
|  |  | Synthesizer |
|  |  | None of these |
| 2 | How is your room currently? | Bright |
|  |  | Dark |
|  |  | Hot |
|  |  | Cold |
|  |  | Fine |
| 3 | From 0 to 10, how much pain are you in? | Zero |
|  | From 0 to 10, how nauseous are you? | One |
|  | From 0 to 10, how happy do you feel? | Two |
|  | From 0 to 10, how stressed are you? | Three |
|  | From 0 to 10, how comfortable are you? | Four |
|  |  | Five |
|  |  | Six |
|  |  | Seven |
|  |  | Eight |
|  |  | Nine |
|  |  | Ten |
| 4 | When do you want me to check back on you? | Today |
|  |  | Tomorrow |

answers with context integration. The primary performance evaluation metric was decoding accuracy rate, which was defined as 1 minus the utterance error rate using the actual and predicted utterances for each prediction type. Here, an utterance refers to one of the question stimuli or answer choices. The utterance error rate was defined as the edit (Levenshtein) distance between the actual and predicted utterance sequences across all test blocks for a participant. This value measures the minimum number of deletions, insertions, and substitutions (at the utterance level) required to convert the predicted utterance sequence into the actual utterance sequence, which is analogous to the word error rate metric commonly used in automatic speech recognition (ASR) systems to assess word-level decoding performance. Thus, the decoding accuracy rate describes the performance of the full decoding approach, including contributions from the speech event detection, utterance classification, and context integration models.

For all participants, accuracy rate for decoding of each prediction type (questions, answers without context, and answers with context) was significantly above chance ($p < 0.05$, one-tailed bootstrap test, 4-way Holm-Bonferroni correction[30]; Fig. 2a for participant 1, Supplementary Fig. 3a for other participants; Supplementary Table 2). Chance accuracy rate was computed using bootstrapped sequences of randomly-sampled utterances (see Section 4.8.3). Overall, the accuracy rates for questions (participant 1: 2.6, participant 2: 3.1, participant 3: 2.1 times the chance level) and answers with context (participant 1: 7.2, participant 2: 3.5, participant 3: 3.7 times the chance level) demonstrate that the full system (event detection, utterance classification, and context integration) achieves reliable decoding of perceived and produced speech from ECoG signals. Importantly, we also observed a significant increase in decoding accuracy rate during answer decoding when context was

integrated compared to when it was not integrated (participant 1: $p = 1.9 \times 10^{-3}$, participant 2: $p = 7.9 \times 10^{-5}$, participant 3: $p = 0.029$, one-tailed permutation test, 4-way Holm-Bonferroni correction). These results indicate that the context integration model was able to leverage the question predictions to improve decoding of the subsequent answer responses for each participant.

To better understand how each of the components contributed to the overall performance of the full system, we examined the utterance classification and context integration models separately from the speech detection model. In this work, we explicitly differentiate between the terms classification and decoding: Given a set of features (such as a time window of neural signals), classification refers to the prediction of a single label from these features, and decoding refers to the prediction of an arbitrary-length label sequence from these features. To evaluate classification performance, we used true event times determined from acoustic transcriptions of the test blocks, ensuring that the appropriate time window of neural signals was associated with each classification target (each test trial). Using these true event times, we calculated question and answer classification accuracy, defined as the proportion of correct utterance classifications in the test blocks. These classification accuracy values directly measure the efficacy of the utterance classifiers and can be compared to the decoding accuracy rates to assess the efficacy of the speech detectors (for an alternative metric, information transfer rate, see Supplementary Note 1 and Supplementary Table 3). For all participants, classification accuracy was above chance for each prediction type ($p < 0.05$, one-tailed bootstrap test, 4-way Holm-Bonferroni correction; Fig. 2b, Supplementary Fig. 3b). Similar to the full system decoding accuracy rate, answer classification accuracy was higher when integrating context (participant 1: $p = 0.033$, participant 2: $p = 1.9 \times 10^{-6}$, participant 3: $p = 9.2 \times 10^{-4}$, one-tailed exact McNemar's test[31], 4-way Holm-Bonferroni correction; see Supplementary Note 2 and Supplementary Table 4 for further characterization of the context integration effects).

We also assessed classification performance using cross entropy, a metric that compares the predicted utterance likelihoods and the actual utterance identities for each trial across all test blocks for a participant (see Section 4.8). Given utterance log likelihoods predicted by a classification model for trials in the test blocks, cross entropy measures the average number of bits required to correctly classify those utterances. These values provide further insight into the performance of the utterance classification and context integration models by considering the predicted probabilities of the utterances (not just which utterance was most likely in each trial). Lower cross entropy indicates better performance. For all participants, cross entropy was better than chance ($p < 0.05$, one-tailed bootstrap test, 4-way Holm-Bonferroni correction; Fig. 2c, Supplementary Fig. 3c) and was significantly better for the answer predictions when integrating context (participant 1: $p = 7.6 \times 10^{-6}$, participant 2: $p = 2.6 \times 10^{-17}$, participant 3: $p = 3.1 \times 10^{-11}$, one-tailed Wilcoxon signed-rank test, 4-way Holm-Bonferroni correction).

To evaluate the performance of the event detector, we computed a detection score that incorporates frame-by-frame detection accuracy and a comparison between the number of detected and actual utterances (Fig. 2d, Supplementary Fig. 3d; see Section 4.8.1). For all participants, detection scores for questions and answers were high (above 85%) but not perfect. This result is consistent with our observation of decoding accuracy rates that were slightly lower than their corresponding classification accuracies.

Finally, to characterize the contribution of individual electrodes during utterance classification and speech detection, we calculated the discriminative power of each ECoG electrode (see

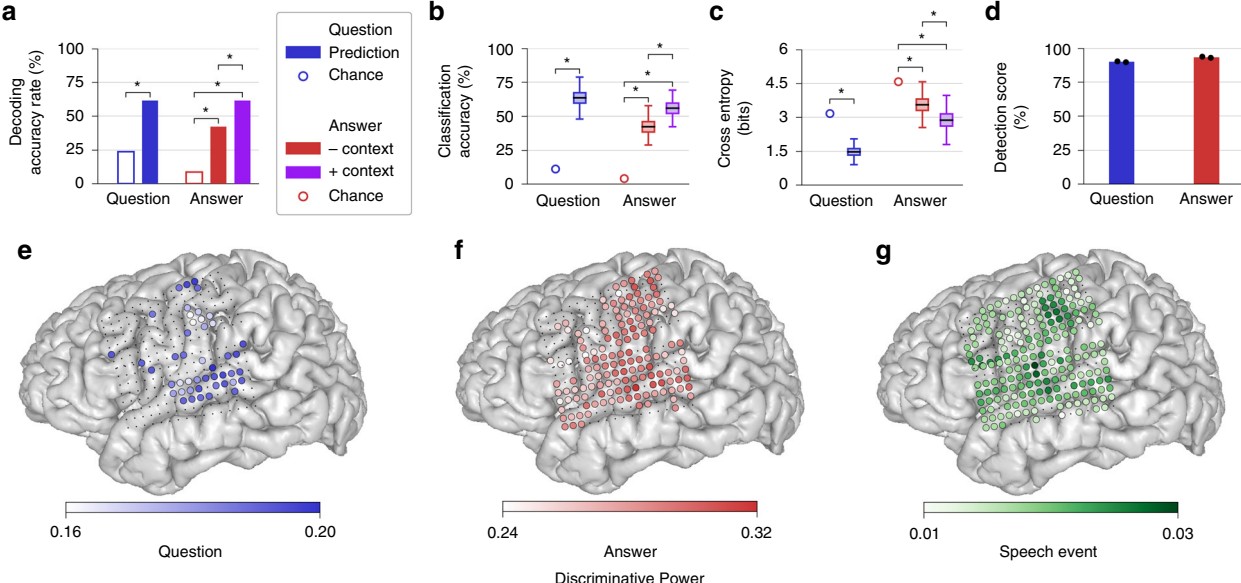

**Fig. 2** Speech decoding and classification results for one participant. **a** Decoding accuracy rate, which measures the full performance of the system, is significantly above chance for questions and answers (without and with context; all *$p < 0.05$, 4-way Holm-Bonferroni correction). Answer decoding accuracy rate is significantly higher with context compared to without context. **b** Classification accuracy (the percent of correctly classified speech events, using true event times) mirrors decoding accuracy rate. (**c**) Cross entropy for utterance classification demonstrates similar patterns of better-than-chance performance and improvement with context (lower values indicate better performance). In (**b**, **c**), the values were computed by bootstrapping across trials. Each boxplot depicts a line marking the median value, box heights representing the interquartile range, and whiskers extending beyond the box edges by 1.5 times the interquartile range. **d** Event detection scores demonstrate near-ceiling performance of the speech detection model for both questions and answers. Black dots depict detection scores on individual test blocks. **e–g** MRI brain reconstructions with electrode locations and discriminative power for each electrode used by (**e**) question, (**f**) answer, and (**g**) speech event discriminative models. Electrodes that were not relevant are depicted as small black dots. See Supplementary Fig. 3 for other participants

Section 4.8.1). Here, discriminative power provides an estimate of how much each electrode contributes to a model's ability to discriminate between utterances or speech events. Although the absolute magnitudes of these values are difficult to interpret in isolation, the spatial distribution of discriminative powers across electrodes indicates which brain areas were most useful for decoding. We found that for question decoding, discriminative power was highest across STG electrodes (Fig. 2e, Supplementary Fig. 3e), which is consistent with auditory responses to heard speech observed in this region. Clusters of discriminative power for question decoding were also observed in vSMC, although the relevant electrodes in this region were sparser and more variable across participants. The electrodes that contributed most to answer decoding were located in both vSMC and STG (Fig. 2f, Supplementary Fig. 3f), reflecting activity related both to speech production and perception of self-produced speech. Lastly, electrodes that contributed to speech detection were distributed throughout sensorimotor and auditory regions (Fig. 2g, Supplementary Fig. 3g).

**Effects of data limitations and hyperparameter selection.** Overall, the reliable decoding performance we observed may reflect certain idiosyncrasies of the neural data and recording constraints associated with each participant. To understand the limitations of the decoding models used in this task, we assessed their performance as a function of several factors that can vary across participants: amount of data used during model fitting, specific model hyperparameters used during testing, and, as described in Supplementary Note 3 and Supplementary Fig. 4, spatial resolution of the cortical signals.

First, we analyzed how the amount of neural data used during training affects decoder performance. For each participant, we fit utterance classification models with neural data recorded during perception and production of an iteratively increasing number of randomly drawn samples (perception or production trials during training blocks) of each utterance. We then evaluated these models on all test block trials for that participant. We found that classification accuracy and cross entropy improved over approximately 10–15 training samples (Fig. 3a, Supplementary Fig. 5a). After this point, performance began to improve more slowly, although it never completely plateaued (except for the answer classifier for participant 2, where 30 training samples were acquired; Supplementary Fig. 5a). These findings suggest that reliable classification performance can be achieved with only 5 min of speech data, but it remains unclear how many training samples would be required before performance no longer improves. We also performed a similar analysis with the detection models to assess speech detection performance as a function of the amount of training data used. We found that detection performance plateaus with about 25% of the available training data (as little as 4 min of data, including silence) for each participant (Supplementary Fig. 6; see Supplementary Method 1 for more details).

Next, we investigated the impact that hyperparameter selection had on classification performance. Hyperparameters are model parameters that are set before training a model on a dataset and are not learned directly from the dataset. Prior to evaluating performance offline with real-time simulations, we performed cross-validated hyperparameter optimization on the models used during decoding (see Section 4.7). Using an iterative optimization algorithm[32,33], we evaluated different sets of hyperparameter

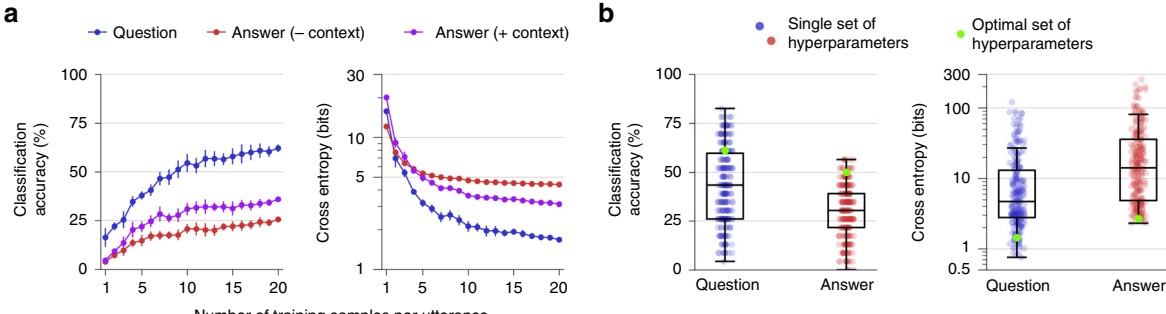

**Fig. 3** Effects of amount of training data and hyperparameter optimization on speech classification for one participant. **a** Classification accuracy and cross entropy as a function of the amount of training data (mean ± s.e.m.). **b** Variability in classification performance across hyperparameter optimization epochs for one test block. Each boxplot depicts a line marking the median value, box heights representing the interquartile range, and whiskers extending beyond the box edges by 1.5 times the interquartile range. Each blue and red dot shows the performance on the test block using a single set of hyperparameters chosen for one epoch during optimization on a separate validation set. Each green dot marks the performance on the test block using the hyperparameters that minimized cross entropy on the validation set (the hyperparameter values used in the main results). See Supplementary Fig. 5 for other participants

values for each test block using a leave-one-block-out cross-validation procedure. We performed 250 optimization epochs for each test block (each epoch evaluated one unique set of hyperparameter values). During the primary performance evaluation for each test block (which were used to obtain the results in Section 2.2), we used the hyperparameter values that produced the best performance on the held-out validation set associated with that block.

To understand how hyperparameter selection affected performance, we compared classification performance on one test block for each participant across the 250 hyperparameter sets that were evaluated for each utterance type (without using the context integration model) during optimization on the associated validation set. For each participant, we observed large variability in classification accuracy and cross entropy across the different hyperparameter sets, suggesting that hyperparameter values can have a large impact on performance (Fig. 3b, Supplementary Fig. 5b). For each participant and metric, we also found that the optimal hyperparameters on the validation set were always better than the median performance observed across all hyperparameter sets. This finding demonstrates that the optimizer successfully chose high-performing hyperparameter values to use during testing and also that hyperparameter values that performed well in certain test blocks are generalizable to other test blocks.

**Viterbi classification and phonetic modeling**. To gain a more intuitive understanding of the neural and stimulus-dependent features that drove decoding performance, we examined the specific phone-level decisions made by the answer classification models (independently from the context integration model) during testing (Fig. 4). These classifiers represented each utterance as a hidden Markov model (HMM), with phones as hidden states and neural data as observed states. During testing, we computed phone likelihoods at each time point during a detected utterance. We then performed Viterbi decoding on the HMM associated with each utterance to compute the most likely path through the hidden states (phones) given the observed sequence of neural data.

We examined how estimated phone likelihoods affected the probability of each utterance across time. For example, when a participant produced the answer "Fine" (in response to the question "How is your room currently?"), an answer classifier used the sequence of phone likelihood estimates (predicted from neural data) to update the predicted probabilities of each possible

answer at each time point during the utterance (Fig. 4a). The pattern of the answer probabilities illustrates how phonetic similarity drives the classifier predictions. For example, the utterances "Fine", "Five", and "Four" remain equally likely until the decoder receives neural activity associated with production of the /ˈaɪ/ phone, at which point "Four" becomes less likely. Subsequently, "Fine" and "Five" are equally likely until the decoder receives neural activity associated with the /n/ phone, at which point "Fine" becomes and remains the most likely utterance. Similarly, there is a brief increase in the probability of "Bright" about halfway through the utterance, consistent with the presence of the /ˈaɪ/ phone (after which the probability decreases). At the end of the utterance, the presence of the /ˈaɪ/ and /n/ phones is associated with an increase in the probability of "Nine".

To understand how much phonetic information the answer classifiers required before finalizing an utterance prediction, for each test trial we computed the earliest time point during Viterbi decoding at which the utterance that was most likely at the end of decoding became and remained more likely than the other utterances. We defined the decision finalization time as the percent of time into the utterance when this time point was reached (using the actual speech onset and offset times from the transcriptions). We computed these decision finalization times for each trial in which the answer classification models correctly predicted the produced answer (94 trials total across all participants and test blocks).

We found that the decision finalization times typically occurred before all of the neural data from an utterance was seen ($p = 2.1 \times 10^{-15}$, one-tailed single-sample Wilcoxon signed-rank test; Fig. 4b). Because some utterances began with the same phones (e.g., the phones /s ˈɪ/ at the start of "Six" and "Synthesizer"), we expected the lower bound for the finalization times to occur after speech onset even if the actual phone identity at each time point was known. To compute this lower bound, we re-calculated the finalization times for these trials using phone likelihoods constructed directly from the phonetic transcriptions. Because no two utterances had the exact same phonetic content, these transcription-based finalization times always occurred before the speech offset ($p = 1.6 \times 10^{-16}$, one-tailed single-sample Wilcoxon signed-rank test). The neural-based finalization times were significantly later than the transcription-based finalization times ($p = 1.2 \times 10^{-10}$, one-tailed Wilcoxon signed-rank test), which is expected when using imperfect phone likelihoods from neural data. Overall, these results demonstrate that the answer

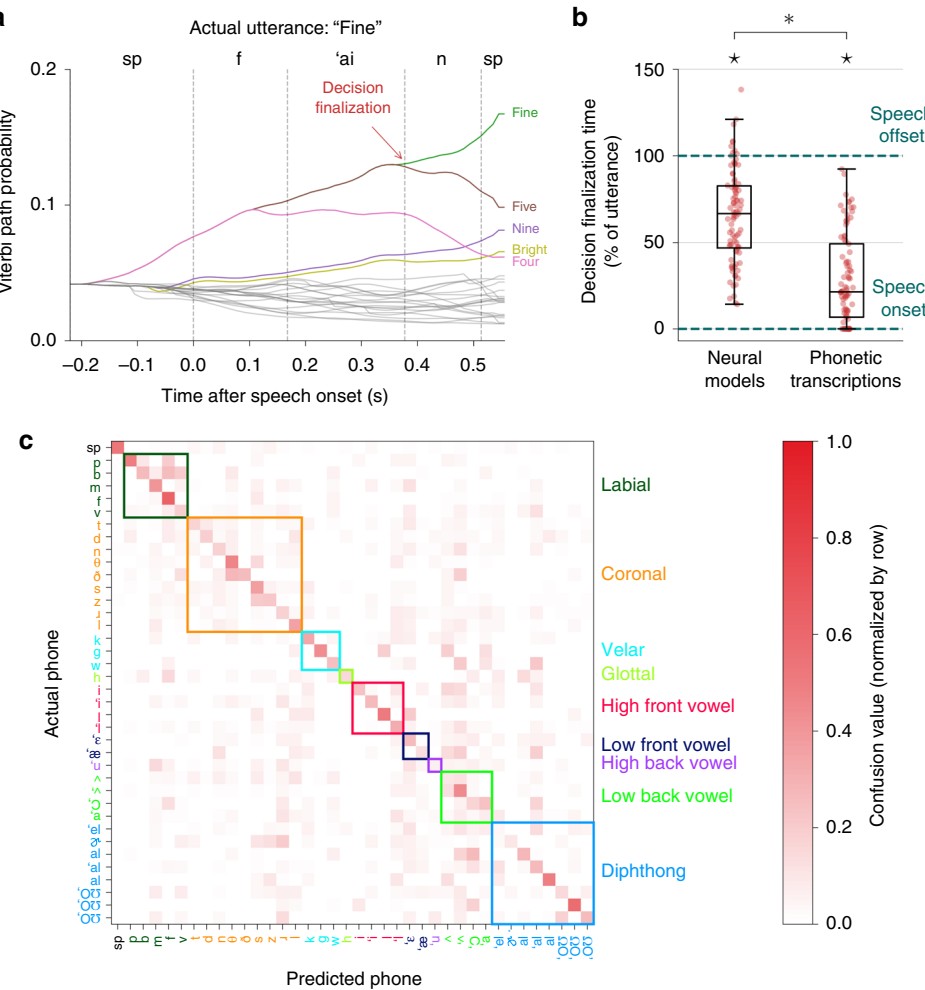

**Fig. 4** Within-trial temporal characteristics and phone-based performance of the answer (speech production) classification models. **a** Viterbi path probabilities during production of the utterance "Fine" demonstrate how the classifier uses phone-level information to predict answers as speech unfolds over time. Each curve depicts the probability of an answer given the neural data at each time point. The probabilities at the final time point represent the answer likelihoods that are passed to the context integration model. Only the five most likely utterances are labeled and colored for visualization purposes. The time at which the correct utterance becomes more likely than the other utterances (and remains more likely throughout the remainder of the decoding window) is marked as the decision finalization time. **b** Decision finalization times for answer classification using neural data and the phonetic transcriptions across all participants and test blocks. Each red dot represents the decision finalization time for a correctly predicted trial (percent of the utterance relative to the actual speech onset and offset for that trial). Each boxplot depicts a line marking the median value, box heights representing the interquartile range, and whiskers extending beyond the box edges by 1.5 times the interquartile range. The observed finalization times typically occurred before speech offset (★$p <$ $10^{-14}$, one-tailed single-sample Wilcoxon signed-rank test), indicating that the classifiers were able to predict the identity of an utterance before processing all time points in the neural (or phonetic) time window associated with an utterance. This characteristic is only partially explained by the stimuli and transcribed vocalizations (*$p < 10^{-9}$, one-tailed Wilcoxon signed-rank test). **c** Phone confusion matrix using the answer phone likelihood model for every time point in each test block across all participants. Colored squares indicate phonetic classes organized by place of articulation. /sp/ is the silence phone. This matrix illustrates reliable discrimination between the majority of the phones and intuitive confusions within articulatory classes (e.g., /s/ vs. /z/)

classifiers were able to finalize classification decisions before the offset of speech using estimated phone likelihoods. Furthermore, this observation cannot be explained entirely by the phonetic separability of the utterances themselves.

We also characterized the performance of the answer phone likelihood models that were used during utterance classification. Across all participants and test blocks, we used the answer phone likelihood models to classify which phone was being produced at each time point. In total, there were 10585 time points that occurred during speech production, and the overall phone classification accuracy across these blocks was 25.12% (the chance level was 2.70% if choosing randomly from the 37 different phones produced during testing). When silence data points were included, the number of time points was 165804 and

the overall phone classification accuracy was 50.97%. This ability of the phone likelihood models to discriminate between phones was a major factor in the success of the utterance classification models during testing.

Based on recent findings suggesting that the clustering of neural encoding of phonemes in vSMC during speech production is largely driven by place of articulation[8,15], we hypothesized that the phone confusions observed in this analysis would be organized by place of articulation. To assess this hypothesis, we divided the set of phone labels into 9 disjoint sets according to place of articulation (excluding the silence token /sp/). We then collapsed each actual and predicted phone label from the phone classification results into one of these 9 phonetic category labels. We found that the mutual information between the actual and

predicted labels using this categorization was significantly higher than randomized phonetic categorizations ($p = 0.0012$, one-tailed bootstrap test)[34], supporting the hypothesis that phone confusions during production can be partially explained by place of articulation. The resulting confusion matrix visually illustrates these findings (Fig. 4c), with a prominent diagonal (indicating good overall classification performance) and confusions that are consistent with this hypothesis (such as the confusions between the alveolar fricatives /s/ and /z/ and between many of the labial phones).

## Discussion

We demonstrate that high-resolution recordings directly from the cortical surface can be used to decode both perceived and produced speech in real-time. By integrating what participants hear and say, we leveraged an interactive question-and-answer behavioral paradigm that can be used in a real-world assistive communication setting. Together, these results represent an important step in the development of a clinically viable speech neuroprosthesis.

The present results provide significant advances over previous work that has used neural signals to decode speech. We used a novel behavioral paradigm that mimics the turn-taking and conversational aspects of natural speech communication. By designing the question/answer sets to contain stimuli that would be challenging and meaningful to decode successfully while leveraging the established functional speech representations in auditory and sensorimotor cortical areas, we were able to evaluate our ability to decode a type of speech that is useful for individuals who could benefit from neuroprosthetic technology. Specifically, conversational speech consists of utterances by both speakers that tend to be related to the same topic. Here, we demonstrated that predicting which question was heard improves the ability to decode the subsequent answer, with the question serving as a constraining context. Also, we did not observe a performance improvement when restricting the possible answer predictions based on the predicted question. Using true question identities as context resulted in increased answer classification accuracy for one participant, suggesting that further improvements to question classification would still be useful in some scenarios. However, this modification did not significantly improve answer predictions for the other two participants, revealing the upper bound of the context integration models. In practice, it may be sufficient to use automatic speech recognition (ASR) with a microphone to decode the questions. However, there are both scientific and technical advantages to a fully-contained and generalizable speech decoding system that uses the listener's perception, including selective attention[35], as context for subsequent decoding.

Our results were achieved with methods that have been used successfully in ASR research and applications[29,36], specifically Viterbi decoding with hidden Markov models (HMMs), except here we used neural activity as features during decoding instead of acoustic signals. We selected an HMM model architecture for several reasons, with arguably the most important being its inherent robustness to certain kinds of variability in the structure of speech. During Viterbi decoding, the answer classifiers were robust to variability in the exact duration and pronunciations of the produced answers because the amount of time each HMM could spend in each phone state was flexible. Similarly, both the question and answer classifiers were robust to slight inaccuracies in the detected speech onsets and offsets because each HMM started and ended with a silence state. The phone likelihood models underlying these utterance classifiers relied on discriminable phonetic encoding in the neural activity, which has been described in previous studies with both perceived[4,12] and

produced[15,16,37] speech. Although other methods such as deep neural network modeling may be able to overcome these and other types of variability, the demonstrated methodologies we used allow for robust decoding of continuous speech from neural activity, including in data-limited settings such as clinical recordings with epilepsy patients.

Additionally, we found that it is both possible and practical to determine which time segments of continuous neural signals are associated with perceived and produced speech events directly from the neural activity itself, similar to previous work on produced speech detection[21]. By training models to detect speech events from neural activity, we were able to achieve reliable detection accuracy for perception and production events even though they occurred intermittently throughout testing blocks that lasted on the order of minutes. Despite the high detection scores exhibited by the speech detectors, however, the observed discrepancies between the decoding accuracy rates and classification accuracies indicate that further improvements to the speech detection approach would improve overall decoder performance.

We identified several participant-specific and practical factors that influenced speech decoding performance. First, it is generally true (in many types of modeling applications) that more training data leads to improved decoder performance. However, we showed here that even in our relatively complex behavioral task involving both speech perception and production, speech detection and utterance classification performance began to plateau after only a few minutes of training data. Although we were limited in how much data we could collect here due to clinical constraints, for patients whose quality of life can be improved by a speech prosthesis, there may be a high tolerance to obtain a substantial amount of speech data to use during training. In those scenarios, alternative models with a greater number of learnable parameters and the ability to model nonlinearities (e.g., artificial neural networks) could leverage additional training data obtained over hours or days[38].

Additionally, the total amount of time that was available with each participant prevented us from using optimized models during online testing. Our results demonstrated that hyperparameter optimization can have a substantial effect on classification performance. Because the discrepancy between hyperparameter values was the only practical difference between the online and offline decoding models, the online decoding performance at the patient's bedside would have been improved to the levels observed in offline real-time analyses if optimization prior to online testing was feasible.

Finally, while we observed that canonical brain networks involved in speech perception and production were engaged in the different phases of the task, there were some differences among participants. With only three participants in this study, we were not able to make quantitative and definitive claims about the relationship between decoder performance and functional-anatomical coverage, although this may be an important factor in determining where to place electrode arrays in patients who will benefit from this technology. Other potential sources of variability in decoder performance across the participants include alertness and task engagement. Additionally, despite using a task that allows participants to choose what to say voluntarily, they were still provided with visual cues listing their response options. While we do not have evidence to suggest that visual presentation or the act of reading influenced the present results, future work could evaluate similar paradigms that do not involve these aspects of visual and language processing.

We also showed that phonetic features were a key driver of classification for produced utterances by characterizing how the

answer classifiers incorporated information across time within individual trials and discriminated between the possible utterances. The HMM-based models learned to recognize neural activity patterns associated with phonetic features (such as coronal articulation and vowel height) and adjusted their online utterance probability estimates depending on the presence of these features at each time point. Consistent with previous findings, the phonetic confusions exhibited by our classifiers were partially explained by place of articulation features[8,15], suggesting that the phone likelihood models struggled to discriminate between within-category speech sounds during decoding. Although these phonetic representations are only an approximation of the underlying kinematic and articulatory representations of speech in vSMC[5,8], the use of simple phonetic labels to describe behavior enabled the classifiers to leverage standard ASR techniques during decoding. Nevertheless, it is likely that decoding performance could be improved by incorporating descriptions of the behavior that are more correlated with the neural representations.

Compared to the HMM-based classifiers used in this work, the state-of-the-art decoding approaches in ASR with acoustic signals are substantially more powerful, leveraging deep and recurrent artificial neural networks to discover structure in the input data[39]. At present, it is difficult to implement these algorithms effectively in neural speech recognition applications that require decoding of speech from neural signals, primarily due to the relatively small amount of data that can be collected to train models[38]. However, advances in data collection, such as chronic recording setups in ambulatory settings[40], and statistical methods for building models using smaller training datasets[41,42] could make these types of sophisticated decoding approaches more practical.

For some impaired individuals, such as patients with locked-in syndrome who are conscious but unable to communicate naturally due to paralysis[22,23,43,44], restoration of limited communicative capability is associated with significant increases in self-reported quality of life[23,44]. Although current state-of-the-art communication prostheses based on letter-by-letter typing, cursor control, and target detection are already beneficial to some patients and are highly generalizable, many are slow and unnatural compared to the type of communication potentially afforded by a speech-based prosthesis, requiring patients to spell out intended messages slowly at rates less than 8 words per minute[25]. An ideal speech prosthesis would be capable of decoding spontaneous, natural speech controlled by a patient's volition and would balance the tradeoff that currently exists in neural prosthetics between generalizability and naturalness. Ultimately, such a system would also generalize to imagined or covertly-produced speech[45], particularly for the case of fully paralyzed individuals. There may be additional challenges in translating the approaches presented here to the imagined speech setting; for example, it is unknown whether phone-based models are appropriate for imagined speech, and the specific training procedures used here may need to be modified in cases where patients are unable to speak or move. Nevertheless, the present results are a promising step towards this goal, demonstrating that produced speech can be detected and decoded from neural activity in real-time while integrating dynamic information from the surrounding context.

## Methods

**Participants**. Three human epilepsy patients undergoing treatment at the UCSF Medical Center participated in this study. For the clinical purpose of localizing seizure foci, ECoG arrays were surgically implanted on the cortical surface of one hemisphere for each participant. All participants were right-handed with left hemisphere language dominance determined by their clinicians.

The research protocol was approved by the UCSF Committee on Human Research. Prior to surgery, each patient gave his or her written informed consent to participate in this research.

**Neural data acquisition**. Participants 1 and 2 were each implanted with two 128-channel ECoG arrays (PMT Corp.) and participant 3 was implanted with a 256-channel ECoG array (Ad-Tech, Corp.). Participants 1 and 3 had left hemisphere coverage and participant 2 had right hemisphere coverage. Each implanted array contained disc electrodes with 1.17 mm exposure diameters arranged in a square lattice formation with a 4 mm center-to-center electrode spacing. We used the open source img_pipe package[46] to generate MRI brain reconstruction images with electrode locations for each participant (Fig. 2, Supplementary Fig. 3).

We used a data acquisition (DAQ) rig to process the local field potentials recorded from these arrays at multiple cortical sites from each participant. These analog ECoG signals were amplified and quantized using a pre-amplifier (PZ5, Tucker-Davis Technologies). We then performed anti-aliasing (low-pass filtering at 1500 Hz) and line noise removal (notch filtering at 60, 120, and 180 Hz) on a digital signal processor (RZ2, Tucker-Davis Technologies). On the DAQ rig, we stored these neural data (at 3051.76 Hz) along with the time-aligned microphone and speaker audio channels (at 24414.06 Hz). These neural data were anti-aliased again (low-pass filtered at 190 Hz) and streamed at a sampling rate of 381.47 Hz to a real-time computer, which was a Linux machine (64-bit Ubuntu 14.04, Intel Core i7-4790K processor, 32 GB of RAM) implementing a custom software package called real-time Neural Speech Recognition (rtNSR)[14].

**High gamma feature extraction**. The rtNSR package implemented a filter chain comprising three processes to measure high gamma activity in real-time (Supplementary Fig. 1). We used high gamma band activity (70–150 Hz) in this work because previous research has shown that activity in this band is correlated with multi-unit firing processes in the cortex[28] and can be used as an effective representation of cortical activity during speech processing[4,10,13–15].

The first of these three processes applied eight band-pass finite impulse response (FIR) filters to the ECoG signals acquired from the DAQ rig (at 381.47 Hz). The logarithmically increasing center frequencies of these filters were 72.0, 79.5, 87.8, 96.9, 107.0, 118.1, 130.4, and 144.0 (in Hz, rounded to the nearest decimal place)[13,15]. The filters each had an order of 150 and were designed using the Parks-McClellan algorithm[47].

The second process in the filter chain estimated the analytic amplitude values for each band and channel using the signals obtained from the band-passing process. An 80th-order FIR filter was designed using the Parks-McClellan algorithm to approximate the Hilbert transform. For each band and channel, this process estimated the analytic signal using the original signal (delayed by 40 samples, which was half of the filter order) as the real component and the FIR Hilbert transform approximation of the original signal as the imaginary component[48]. The analytic amplitudes were then computed as the magnitudes of these analytic signals. This filtering approach was applied to every fourth sample of the received signals, effectively decimating the signals to 95.37 Hz.

The final process in the filter chain averaged analytic amplitude values across the eight bands, yielding a single high gamma analytic amplitude measure for each channel.

After filtering, the high gamma signals were z-scored using Welford's method with a 30-second sliding window[49]. To mitigate signal artifacts such as channel noise and epileptic activity, we clipped the z-score values to lie within the range of [−3.5, 3.5]. We used the resulting z-scores as the representation of high gamma activity in all subsequent analyses and real-time testing.

**Experimental task design**. The overall goal of this task was to demonstrate real-time decoding of perceived and produced speech while leveraging contextual relationships between the content of the two speech modalities. To achieve this, we designed a question-and-answer task in which participants listen to questions and respond verbally to each question with an answer. There were 9 pre-recorded acoustic question stimuli and 24 possible answers (Table 1). Questions were recorded by a female speaker at 44.1 kHz and were presented to each participant aurally via loudspeakers. Each visual answer choice was represented as a small rectangle containing the text prompt and a small image depicting the text (Fig. 1b; images were included to increase participant engagement). The stimuli were divided into four question/answer sets (QA sets 1–4). The answers in each QA set represented the answer choices that would appear on the screen for each of the questions in that set.

We used the following three types of task blocks: (1) question (perception) training, in which participants heard each question 10 times in a random order (stimulus length varied from 1.38–2.42 s in duration with an onset-to-onset interval of 3 s); (2) answer (production) training, in which participants read each possible answer choice aloud 10 times in a random order (each answer appeared on the screen with a gray background for 0.5 s, was changed to a green background for 1.5 s to represent a go cue for the participant to read the answer, and removed from the screen for 0.5 s before the next answer was displayed); and (3) testing, in which participants heard questions and responded verbally with answers (choosing a response from the possible options presented on the screen after each question).

During the testing blocks, a green circle appeared on the screen after each question was presented to cue participants to respond aloud with an answer of their choosing. We encouraged the participants to choose different answers when they encountered the same questions, although they were free to respond with any of the presented answer choices during each trial. There was 2–3 s of silence and a blank screen between each trial. In each block, the questions played to the participant were chosen based on how many questions and answers are in each QA set (questions with more valid answers had a greater chance of being played in each trial). Trials in which the participant failed to respond or responded with an invalid choice (less than 0.5% of trials) were excluded from further analysis. There were 26 question-and-answer trials in each testing block.

During each block, time-aligned behavioral and neural data were collected and stored. The data collected during training blocks were used to fit the decoding models. The data collected during testing blocks were used to decode the perceived questions and produced answers in real-time and were also used offline during hyperparameter optimization.

**Phonetic transcription**. After data collection, both questions and answers were phonetically transcribed from the time-aligned audio using the p2fa package[50], which uses the Hidden Markov Model Toolkit and the Carnegie Mellon University Pronouncing Dictionary[51,52]. The phone boundaries were manually fine-tuned using the Praat software package[53]. Including a silence phone token /sp/, there were a total of 38 unique phones in the question stimuli and 38 unique phones in the produced answer utterances, although these two phone sets were not identical.

**Modeling**. After collecting training data for a participant, we fit models using the time-aligned high gamma z-score neural data and phonetic transcriptions. Model fitting was performed offline, and the trained models were saved to the real-time computer to be used during online testing. As described in Section 4.7, the values for many model parameters that were not learned directly from the training data were set using hyperparameter optimization. We used three types of models in this work: speech detection models, utterance classification models, and context integration models.

*Speech detection*: Before using the neural data to train speech detection models, we analyzed the collected data to identify electrodes that were responsive to speech events[13,14]. For each time point in the neural data, we used the phonetic transcriptions to determine if that time point occurred during speech perception, speech production, or silence. We then performed Welch's analysis of variance (ANOVA) on each electrode to identify channels that were significantly modulated by the different types of speech events. Channels that had a Welch's ANOVA p-value less than a threshold hyperparameter were included in the feature vectors used to train and test the speech detection models.

Speech events were modeled discriminatively as conditional probability distributions of the form $p(h_t|y_t)$. Here, $h_t$ represents the speech event at time $t$ and is one of the values in the set {perception,production,silence}, and $y_t$ is the spatiotemporal neural feature vector at time $t$. The $h_t$ labels were determined from the phonetic transcriptions: for any given time index $t$, $h_t$ was perception if the participant was listening to a phone at time $t$, production if the participant was producing a phone at time $t$, or silence otherwise. Each of these feature vectors was constructed by concatenating high gamma z-score values for relevant electrodes across all of the time points in a time window relative to the target time point, capturing both spatial (multiple electrodes) and temporal (multiple time points) dynamics of the cortical activity[13,14] (Supplementary Fig. 7). Specifically, a feature vector associated with the speech event label at some time index $t$ consisted of the neural data at the time indices within the closed interval $[t + v_{\text{shift}}, t + v_{\text{shift}} + v_{\text{duration}}]$, where $v_{\text{shift}}$ and $v_{\text{duration}}$ represent the window onset shift and window duration, respectively, and were determined using hyperparameter optimization.

To compute the speech event probabilities $p(h_t|y_t)$ at each time point, we fit a principal component analysis (PCA) model with the constraint that the dimensionality of the projected feature vectors would be reduced to the minimum number of principal components required to explain a certain fraction of the variance across the features (this fraction was a hyperparameter determined during optimization). We then used these new projected feature vectors and the speech event labels to fit a linear discriminant analysis (LDA) model implementing the least-squares solution with automatic shrinkage described by the Ledoit-Wolf lemma[54]. After training, these PCA-LDA models could be used during testing to extract the principal components from a previously unseen spatiotemporal feature vector and predict speech event probabilities from the resulting projection (the LDA model assumed flat class priors when computing these probabilities). We used the Python package scikit-learn to implement the PCA and LDA models[55].

During testing, the predicted speech event probabilities were used to detect the onsets and offsets of speech events (Supplementary Fig. 2) with a multi-step approach. For every time point $t$, the $p(h_t|y_t)$ probabilities were computed using the speech event probability model (Supplementary Fig. 2a). For perception and production, these probabilities were smoothed using a sliding window average (Supplementary Fig. 2b). Next, these smoothed probabilities were discretized to be either 1 if the detection model assigned time point $t$ to the associated speech event type or 0 otherwise (Supplementary Fig. 2c). These probability-thresholded binary values were then thresholded in time (debounced); a speech onset (or offset) was

only detected if this binary value changed from 0 to 1 and remained 1 for a certain number of time points (or the opposite for offsets; Supplementary Fig. 2d). Whenever a speech event offset was detected (which could only occur after an onset had been detected), the neural data in the detected window were passed to the appropriate utterance classification model (Supplementary Fig. 2e). The number of recent time points used during probability averaging, probability threshold value, time threshold duration, and onset and offset index shifts (integers added to the predicted onset and offset time indices before segmenting the neural data) were all treated as hyperparameters and set via optimization (with separate parameters for perception and production).

*Utterance classification*: For each participant and utterance type (questions and answers), we used classification models to predict the likelihood of each utterance given a detected time segment of neural activity. For each utterance, we constructed a hidden Markov model (HMM) to represent that utterance[56], with phones $q_t$ as hidden states and spatiotemporal neural feature vectors $y_t$ as observed states at each time index $t$. Each of these HMMs was created using the representative phone sequence for the associated utterance (determined from the phonetic transcriptions). The transition matrix for each HMM, which specified the transition probabilities $p(q_{t+1}|q_t)$, was defined such that each hidden state was one of the phones in the associated representative sequence and could only self-transition (with some probability $p_{\text{self}}$) or transition to the next phone in the sequence (with probability $1 - p_{\text{self}}$). A self-transition probability of 1 was used for the final state. We used the silence phone token /sp/ as the initial and final states for each HMM. Given a time series of high gamma z-score values, each of these HMMs yielded the likelihood of observing those neural features during perception or production of the underlying phone sequence. These likelihoods are robust to natural variability in the durations of the phones in the sequence, which is a key motivation for using HMMs in this approach (even with a single speaker producing the same utterance multiple times, phone durations will vary).

Similar to the relevant electrode selection procedure used for the speech detection models, we identified which channels should be considered relevant to the type of speech processing associated with each utterance type. Using the three previously described data subsets (perception, production, and silence), we performed two-tailed Welch's t-tests for each channel between the appropriate subsets for each utterance type (perception vs. silence for questions and production vs. silence for answers). Channels with a p-value less than a threshold hyperparameter value were considered relevant for the current utterance type and were used during subsequent phone likelihood modeling.

PCA-LDA models were then trained to compute the phone emission likelihoods $p(y_t|q_t)$ at each time point $t$. The hyperparameters associated with these models, including the feature time window parameters and the PCA minimum variance fraction, were optimized separately from the parameters in the speech event model.

During testing, we used Viterbi decoding on each HMM to determine the likelihood of each utterance given a detected time segment of high gamma z-scores[13,14,29,45] (Supplementary Fig. 8). We computed the log likelihood of each utterance using the following recursive formula:

$$v_{(t,s)} = w_e \log p(y_t|s) + \max_{i \in S} \left[ v_{(t-1,i)} + \log p(s|i) \right], \tag{1}$$

where $v_{(t,s)}$ is the log probability of the most likely Viterbi path that ends in phone (state) $s$ at time $t$, $p(y_t|s)$ is the phone emission likelihood (the probability of observing the neural feature vector $y_t$ if the current phone is $s$), $p(s|i)$ is the phone transition probability (the probability of transitioning from phone $i$ to phone $s$), $w_e$ is an emission probability scaling factor (a model hyperparameter) to control the weight of the emission probabilities relative to the transition probabilities (see Supplementary Note 4), and $S$ is the set of all possible phones. To initialize the recursion, we forced each Viterbi decoding procedure to start with a Viterbi path log probability of zero for the first state (the initial silence phone /sp/) and negative infinity for every other state.

After decoding for each HMM, the Viterbi path log probability at the final state and time point for that HMM represents the log likelihood $\ell_u$ of the corresponding utterance $u$ given the neural data. Log probabilities are used here and in later computations for numerical stability and computational efficiency.

The computed log likelihoods for each utterance were then smoothed and normalized using the following formula:

$$\ell_u^* := \omega \ell_u - \log \left[ \sum_{j \in U} \exp(\omega \ell_j) \right], \tag{2}$$

where $\ell_u^*$ is the smoothed and normalized log likelihood for utterance $u$, $\omega$ is the smoothing hyperparameter, and $U$ is the set of all valid utterances (for the current utterance type). Because differences in utterance log likelihoods can be large (e.g., in the hundreds), the smoothing hyperparameter, which lay in the closed interval [0, 1], was included to allow the model to control how confident its likelihood predictions were. The closer $\omega$ is to zero, the smoother the log likelihoods are (less sample variance among the log likelihoods). The final log term in Eq. 2 represents the LogSumExp function and was used to compute the normalization constant for the current smoothed log likelihoods. After computing this constant and subtracting it from the smoothed log likelihoods, the $\ell_u^*$ values satisfied the

following equality:

$$\sum_{j \in U} \exp(\ell_j^*) = 1. \tag{3}$$

These $\ell_u^*$ values were used as the utterance classification model's estimate of the utterance log likelihoods given the corresponding neural data.

*Context integration*: Because each answer was only valid for specific questions and an answer always followed each question, we developed a context integration model that used predicted question likelihoods to update the predicted answer probabilities during testing. Based on our previous demonstration that auditory sentences could be decoded from neural activity[14], we hypothesized that we could use reliable decoding of the questions to improve answer predictions.

Prior to testing, we defined the relationships between questions and answers in the form of conditional probabilities. These probabilities, referred to as the context priors, were computed using the following formula:

$$p(u_a | u_q) = \begin{cases} \frac{1}{N_{A,q}} & \text{if } u_a \text{ and } u_q \text{ are in same QA set,} \\ 0 & \text{otherwise,} \end{cases} \tag{4}$$

where $p(u_a | u_q)$ is the context prior specifying the probability of responding to the question $u_q$ with the answer $u_a$ and $N_{A,q}$ is the number of answers in the same question-and-answer (QA) set as $u_q$ (the number of valid answers to $u_q$; Table 1). These context priors assume that the valid answers to any question are equally likely.

During testing, the context integration model receives predicted utterance log likelihoods from both the question and answer classification models. Each time the model receives predicted question log likelihoods (denoted $\ell_{U_Q}^*$, containing the log likelihoods $\ell_{u_q}^*$ for each question utterance $u_q$), it computes prior log probabilities for the answer utterances from these question likelihoods and the pre-defined context priors using the following formula:

$$\log p_Q(u_a) = \log \left\{ \sum_{u_q \in U_Q} \exp \left[ \log p(u_a | u_q) + \ell_{u_q}^* \right] \right\} + c, \tag{5}$$

where $p_Q(u_a)$ is defined as the prior probability of the answer utterance $u_a$ computed using $\ell_{U_Q}^*$, $U_Q$ is the set of all question utterances, and $c$ is a real-valued constant. Each time the model receives predicted answer log likelihoods (the $\ell_{u_a}^*$ values for each answer utterance $u_a$), it computes posterior log probabilities for the answer utterances from these answer likelihoods and the answer priors. The unnormalized log posterior probabilities $\phi_{u_a}$ were computed for each answer utterance $u_a$ using the following formula:

$$\phi_{u_a} := m \log p_Q(u_a) + \ell_{u_a}^* + d, \tag{6}$$

where $m$ is the context prior scaling factor and $d$ is a real-valued constant. Here, $m$ is a hyperparameter that controls the weight of the answer priors relative to the answer likelihoods (a larger $m$ causes the context to have a larger impact on the answer posteriors). We then normalize these answer log posterior values using the following formula:

$$\phi_{u_a}^* := \phi_{u_a} - \log \left[ \sum_{j \in U_A} \exp(\phi_j) \right], \tag{7}$$

where $\phi_{u_a}^*$ is the normalized log posterior probability of $u_a$ and $U_A$ is the set of all answer utterances. The constants $c$ and $d$ do not need to be computed in practice because they are canceled out during the normalization step in Eq. 7. These $\phi_{u_a}^*$ values satisfy the following equality:

$$\sum_{j \in U_A} \exp(\phi_j^*) = 1. \tag{8}$$

Finally, the predicted utterance identities are computed as:

$$\hat{u}_q = \underset{u_q \in U_Q}{\operatorname{argmax}} \ell_{u_q}^*, \tag{9}$$

$$\hat{u}_{a-} = \underset{u_a \in U_A}{\operatorname{argmax}} \ell_{u_a}^*, \tag{10}$$

$$\hat{u}_{a+} = \underset{u_a \in U_A}{\operatorname{argmax}} \phi_{u_a}^*, \tag{11}$$

where $\hat{u}_q$, $\hat{u}_{a-}$, and $\hat{u}_{a+}$ are the system's predictions for questions, answers without context, and answers with context, respectively. The $\hat{u}_q$ and $\hat{u}_{a+}$ predictions are the system outputs during decoding, and the $\hat{u}_{a-}$ predictions are used in offline analyses. For a more thorough mathematical description of the context integration procedure, see Supplementary Note 5.

Although an answer followed each question during testing, it was possible for the speech detector to fail to detect question or answer events (or to detect false positives). Because of this, we did not force the context integration model to always expect answer likelihoods after receiving question likelihoods or vice versa. Instead, during each test block, we maintained a set of values for the answer priors that were

only updated when a new set of question likelihoods was received. When a new set of answer likelihoods was received, the current answer prior values were used to compute the posteriors. If answer likelihoods were received before receiving any question likelihoods, answer posteriors and answer with context predictions would not be computed from those likelihoods (although this did not actually occur in any test blocks).

**Hyperparameter optimization.** Each type of model (speech detection, utterance classification, and context integration) had one or more parameters that could not be learned directly from the training data. Examples of physiologically relevant hyperparameters include a temporal offset shift between perceived and produced phones and the neural data (which could account for neural response delays or speech production planning), the duration of the spatiotemporal neural feature vectors used during model training and testing, and a *p*-value threshold used when deciding which electrodes should be considered relevant and included in the analyses.

Instead of manually selecting values for these hyperparameters, we performed cross-validated hyperparameter optimization using the hyperopt Python package[32,33]. This package uses a Bayesian-based optimization algorithm called the Tree-structured Parzen Estimator to explore a hyperparameter space across multiple epochs. Briefly, this optimization approach samples hyperparameter values from pre-defined prior distributions, uses a loss function to evaluate the current hyperparameters, and then repeats these steps using knowledge gained from the evaluations it has already performed. After a desired number of epochs, the hyperparameter set associated with the minimal loss value across all epochs is chosen as the optimal hyperparameter set.

We performed hyperparameter optimization for each participant, model type, and test block. We used a leave-one-block-out cross-validation scheme for each test block. Specifically, during an optimization run for any given test block, the hyperparameters were evaluated on a held-out validation set comprising all of the other test blocks available for the current participant. We used 250 epochs for each optimization run. All of the hyperparameters that were set via optimization are described in Supplementary Table 5, and the full optimization procedure is described in Supplementary Method 2.

**Evaluation methods and statistical analyses.** *Primary evaluation metrics*: We used the following metrics during the primary evaluations of our system: decoding accuracy rate, classification accuracy, cross entropy, speech detection score, and electrode discriminative power (Fig. 2). The decoding accuracy rate metric represented the full performance of the system (the combined performance of the speech detection, utterance classification, and context integration models). When computing the accuracy rates for each prediction type (questions, answers without context, and answers with context) and participant, we obtained overall actual and predicted sequences by concatenating the actual and predicted utterances across all of the test blocks. We then calculated an utterance error rate using these sequences, which is an analog of the commonly used word error rate metric and is a measure of the edit (Levenshtein) distance between the actual and decoded utterance label sequences in a given test block. The accuracy rate was then computed as 1 minus the utterance error rate (or 0 if this difference would be negative, although this was never observed in our experiments).

Classification accuracy and cross entropy metrics were computed for each participant by using only the utterance classification and context integration models (and not the speech detection models). In this approach, we performed decoding on the test blocks using the actual speech event times and the previously trained utterance classification models. Because the HMMs used to represent the utterances were designed to start and end the Viterbi decoding process during silence, we padded 300 ms of silence time points before and after the utterance in each speech-related time window of neural data passed to the classifiers. We then performed context integration model optimization with these new classification results and applied the optimized context integration models to the results. After this step, we pooled all of the pairs of actual and predicted utterance labels for each prediction type across all of the test blocks for each participant.

Classification accuracy was defined as the proportion of trials in which the utterance classification model correctly predicted the identity of the utterance. To obtain the mean and variance of the classification accuracy, we used classification accuracies computed on bootstrapped resamples of the trials (one million resamples). To measure information transfer rate (which we did not consider a primary evaluation metric in this work), we used these classification accuracy values, speech durations from the test blocks, and the number of possible answer responses (see Supplementary Note 1 for details).

The cross entropy metric quantified the amount of predictive information provided by the utterance classification and context integration models during testing and hyperparameter optimization. We computed cross entropies using the surprisal values for each classification trial, prediction type, and participant. For a given trial and prediction type, the relevant surprisal value for that trial is equal to the negative of the predicted log probability associated with the actual utterance label. The cross entropy is equal to the mean of these surprisal values. To obtain the mean and variance of the cross entropy, we used cross entropies computed on

bootstrapped resamples of the trials (one million resamples). Lower cross entropy indicates better performance.

To evaluate and optimize the speech detector, we created a score metric that computes a weighted combination of a frame-by-frame accuracy $a_{\text{frame}}$ and a general event detection accuracy $a_{\text{event}}$. The frame-by-frame accuracy measures the performance of the speech detector using the detected presence or absence of a speech event at each time point. This measure is analogous to sensitivity and specificity analyses commonly used for binary prediction. Phonetic transcriptions were used to determine the actual times of the speech events and compute true positives, true negatives, false positives, and false negatives. When using these transcribed speech times, we decremented each speech onset time and incremented each speech offset time by 300 ms to label some silence time points before and after each utterance as positive frames. We performed this modification to encourage the optimizer to select hyperparameters that would include silence before and after each utterance in the detected neural feature time windows, which is useful during utterance classification. We calculated the frame-by-frame accuracy measure using the following formula:

$$a_{\text{frame}} := \frac{w_{\text{P}}N_{\text{TP}} + (1 - w_{\text{P}})N_{\text{TN}}}{w_{\text{P}}N_{\text{P}} + (1 - w_{\text{P}})N_{\text{N}}}, \qquad (12)$$

where $w_{\text{P}}$ is the positive weight fraction, $N_{\text{TP}}$ is the number of true positives detected, $N_{\text{TN}}$ is the number of true negatives detected, $N_{\text{P}}$ is the total number of positive frames in the test data, and $N_{\text{N}}$ is the total number of negative frames in the test data. The positive weight fraction was included to allow control over how important true positive detection was relative to true negative detection. In practice, we used $w_{\text{P}} = 0.75$, meaning that correctly detecting positive frames was three times as important as correctly detecting negative frames. We used this value to encourage the optimizer to select hyperparameters that would prefer to make more false positive errors than false negative errors, since the performance of the utterance classifiers should diminish more if a few speech-relevant time points were excluded from the detected time window than if a few extra silence time points were included. The general event detection accuracy, which measures how well the speech events were detected without considering which time points were associated with each event, was computed using the following formula:

$$a_{\text{event}} := 1 - \min\left(1, \frac{|N_{\text{DE}} - N_{\text{AE}}|}{N_{\text{AE}}}\right), \qquad (13)$$

where $N_{\text{DE}}$ and $N_{\text{AE}}$ are the number of detected and actual speech events in the current test block, respectively. To compute the speech detection score $s_{\text{detection}}$, these two measures were combined using the following formula:

$$s_{\text{detection}} = w_{\text{F}}a_{\text{frame}} + (1 - w_{\text{F}})a_{\text{event}}, \qquad (14)$$

where $w_{\text{F}}$ is the frame-by-frame accuracy weight fraction, which allows control over how much impact the frame-by-frame accuracy measure has on the speech detection score relative to the general event detection accuracy. In practice, we let $w_{\text{F}} = 0.5$ for an equal weighting between the two measures. For each participant and utterance type, the overall detection score was computed by taking the average of the detection scores for each test block.

To assess the importance of each electrode during phone and speech event likelihood modeling, we estimated the discriminative power of each electrode within the trained PCA-LDA models[13]. We arbitrarily selected a test block for each participant and obtained the trained and optimized utterance classification and speech detection models associated with that test block. For each of these models, we examined the learned parameters within the LDA model. For each feature in the LDA model (which is a principal component), we measured the between-class variance for that feature by computing the variance of the corresponding class means. We used the values along the diagonal of the shared covariance matrix as a measure of the within-class variance of each feature (because we did not force diagonal covariance matrices in the LDA models, this is only an approximation of the true within-class variances). Similar to a coefficient of determination ($R^2$) calculation, we then estimated the discriminative power for each LDA feature as a ratio of the between-class variance to the total variance using the following formula:

$$\eta_i = \frac{\sigma_{\text{b},i}^2}{\sigma_{\text{w},i}^2 + \sigma_{\text{b},i}^2}, \qquad (15)$$

where $\eta_i$, $\sigma_{\text{b},i}^2$, and $\sigma_{\text{w},i}^2$ are the estimated discriminative power, between-class variance, and within-class variance, respectively, for the $i$th LDA feature. To obtain the discriminative powers for each original feature in the spatiotemporal neural feature vectors (the inputs to the PCA model), the absolute values of the PCA component weights were used to project the LDA feature discriminative powers back into the original feature space. Finally, the discriminative power for each electrode was set equal to the maximum discriminative power value observed among the original features associated with that electrode (that is, the maximum function was used to aggregate the discriminative powers across time for each electrode within the spatiotemporal feature vectors). The resulting discriminative power values were used to quantify the relative contributions of each electrode during phone or speech event discrimination.

*Auxiliary decoding analyses*: As described in Section 2.3, we investigated the sensitivity of the decoding models to limited data availability and sub-optimal

hyperparameter configurations (Fig. 3). Thorough descriptions of these analyses are provided in Supplementary Method 1. The analysis of how spatial resolution affected decoder performance is described in Supplementary Note 3.

As described in Section 2.4, we performed additional analyses on the Viterbi decoding and phone likelihood modeling approaches used by the answer classifiers (Fig. 4). Thorough descriptions of these analyses are provided in Supplementary Method 3.

When performing answer classification with hard or true priors instead of soft priors, the question likelihoods in each trial were modified prior to context integration. For hard priors, the likelihood of the most likely question was set to 1 and the likelihoods of the other questions were set to 0. For true priors, the likelihood of the question that was actually presented to the participant was set to 1 and the likelihoods of the other questions were set to 0. After this modification, the context integration procedure was performed normally to obtain the answer predictions.

*Statistical testing*: The statistical tests used in this work are all described in Section 2. For all tests, we considered p-values less than 0.05 as significant. We used a 4-way Holm-Bonferroni correction[30] for the chance comparisons with the three prediction types (questions, answers without context, and answers with context) and the answer with vs. without context comparison because the neural data used during these analyses were not independent of each other. Thorough descriptions of all of the statistical tests are provided in Supplementary Method 4.

**Real-time decoding**. In our previous work, we introduced the rtNSR software package[14]. Written in Python[57], this package is flexible and efficient due to its modular structure and utilization of software pipelining[58]. With further development, we used rtNSR here to present the audio and visual stimuli, process the neural signals, and perform speech decoding in real-time (Supplementary Fig. 9). We also used it for offline model training and data analysis.

Due to clinical time constraints, we were not able to perform hyperparameter optimization prior to real-time testing with the participants. All of the results reported in this work were computed using offline simulations of the data with the rtNSR system, a process that we described in our previous work. During the offline simulations, the real-time process that reads samples from the real-time interface card is replaced with a process that simulates input samples from a dataset on disk. The remainder of the decoding pipeline remains the same. During online testing at the patient's bedside, the system performed decoding without experiencing systematic/runtime errors and with negligible latency using hyperparameter values chosen via trial and error on datasets that were previously collected. Therefore, we can reasonably expect that the decoding results we observe in our offline simulations would have been identical to those in the online setting with the patients, since the only relevant differences between the online and offline tests were the specific values of the hyperparameters.

**Reporting summary**. Further information on research design is available in the Nature Research Reporting Summary linked to this article.

## Data availability
Deidentified copies of the data used in these analyses can be provided upon reasonable request. Please contact the corresponding author via email with any inquiries about the data.

## Code availability
The rtNSR software package used for real-time demonstration and offline analyses can be provided upon reasonable request. Please contact the corresponding author via email with any inquiries about the code.

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

## Acknowledgements

We thank Gopala Anumanchipalli for feedback on the utterance classification methodology, Garret Kurteff for help with aesthetic design of the answer stimuli and for phonetic transcriptions, David Xie and Ryon Sabouni for additional help with the phonetic transcriptions, and Emily Mugler for feedback throughout various stages of the project. The authors also thank the members of E.F.C.'s lab for help during data recording and the patients who volunteered to participate in this work. This work was funded by a research contract under Facebook's Sponsored Academic Research Agreement. The images included in the visual representations of the answer utterances (displayed to participants during training and testing blocks) were made by www.freepik.com from www.flaticon.com.

## Author Contributions

D.A.M. developed the rtNSR system and performed all the data collection and analyses. M.K.L. provided project guidance and edited the Supplementary Movie. J.G.M. advised on various mathematical and statistical aspects of the project. E.F.C. led the research project. D.A.M. and M.K.L. wrote the manuscript with input from all authors.

**Additional information**

**Competing interests:** The authors declare no conflicts of interest.

