## [Peer Review File · Nature Communications]

Reviewers' Comments:

Reviewer #1:

Remarks to the Author:

This manuscript investigates real-time speech decoding using electrocorticographic (ECoG) signals recorded from the lateral surface of the human cortex. For this purpose, 3 human subjects with intractable epilepsy were implanted with high-density electrodes over their left or right hemisphere. The subjects performed a questions-and-answer task in which they uttered answers to a set of 9 randomly and repetitively presented questions. A real-time system acquired the ECoG signals, and, in a multi-step process, detected and classified questions and answers from the neural data alone. The presented results show that this system was able to classify the questions and answers well above chance level. The presented post-hoc analysis shed some light on the aspects that govern this classification accuracy.

The presented manuscript investigates a relevant question, i.e., whether previously presented offline-based speech decoding results can be translated into a viable real-time system. For this, the authors utilize sophisticated recording, analysis and decoding techniques. With this, the presented manuscript present an impressive demonstration of real-time speech decoding from neural signals. This could be seen as a major step towards a viable speech prosthesis. That being said, the manuscript could improve in providing significant additional neuroscientific insight beyond that relevant for the technical implementation of such a speech prosthesis. In addition, there are several aspects in the presentation of the manuscript that could be improved. Specifically, I would like the authors to address the major issues listed below:

Major Issues:

- (1) The Introduction could be improved by presenting the scientific and technological aspects that that need to be investigated to realize the real-time decoding system. This could provide a parallel structure for the respective methods and results.
- (2) The presented results in Figures 2 and S3 could be improved by overlaying functional mapping results to better understand the relevance of specific functional areas in the decoding process.
- (3) The presented results in Figure 3 could be split into the individual steps of the detection and classification process. This could be a supplementary figure that sheds some light on which aspect of this multi-step process requires more training data. It would also make it easier to understand what the accuracies of each of these steps are.
- (4) It would be interesting to understand whether such high spatial electrode density is actually necessary for a speech prosthesis. Perhaps this could be done by averaging across electrodes, which could effectively simulate a less spatially dense grid. The results of this classification simulation could be presented in a figure not unlike Figure 3.
- (5) The number of trials for each subject, and the duration during which they were recorded is not clearly presented in the manuscript.
- (6) The variability in the utterance could be a limiting factor that warrants a more sophisticated analysis.
- (7) The context integration is necessary for the real-time system. It should be clearly stated that this is only used for the real-time and not for the post-hoc analyses.

(8) Section 4.8.1. mentions that "Classification accuracy and cross entropy metrics were computed for each participant by using only the utterance classification and context integration models (and not the speech detection models)." This should be mentioned upfront in the results.

(9) The Discussion could improve by presenting the necessary scientific and technological steps to translate this demonstration into viable speech prosthesis for locked-in patients. For example, how would the lack of acoustic feedback, and the lack of initial training data affect this approach? What are the remaining scientific and technical hurdles? Which of the many steps in the detection and classification process clearly require improvement?

Reviewer #2:

Remarks to the Author:

The authors propose a novel speech prosthesis that integrates recognition of auditory questions to provide a context for decoding spoken answers. They test this concept using a discrete question and answer set. The study is timely and important. Overall, the study is well executed and the manuscript is clearly written. There are a few areas where I believe descriptions could be improved and claims should be dialed in to more appropriately place this work within the current state of the field.

1) In a few places in the text, the authors use the phrase "high-accuracy rate." I suggest that the authors dial back the use of the term "high-accuracy" unless they can provide clear justification and apple-to-apples quantitative comparison to the current state of the art.

2) The authors refer to the task as "naturalistic," however as described in the methods and in figure 1, the task is externally cued with respect to timing and answers are constrained. While the work conceptually pushes towards naturalistic usage, the study design and the context in which the methods are tested are not naturalistic. This language should be dialed back and the difference between the task and a naturalistic setting should be discussed.

3) There is a high degree of variability in performance across subjects. While I respect the authors statement that it is not possible to make definitive claims about this variability, the paper would be strengthened by elaborating on potential sources of variability in the discussion.

4) The authors very responsibly present the detection scores as an aggregate of detection, context integration, and utterance classification. However, it would be helpful to further understand how failures in detection and context (question) identification degrade performance — the authors could compute performance assuming perfect detection and/or context identification. In the case of context identification, performance under hard assignment of context could provide a useful comparison point to the likelihood based soft assignment. This hard assignment could be through the decoder (which would allow us to assess the value of soft assignment) or via an oracle (cheating by using the actual context as presented).

5) Although the supplement is very complete in its description of hyperparameters, the results section describing hyperparameters could be strengthened by providing a concise description of the physiologically relevant hyperparameters (e.g. parameters control time delays, electrode selection). The authors show the importance of hyperparameters, but showing a high degree of variability in performance. Related to this, the authors should provide a description of the stability of tuned hyperparameters as a function of fold — as this stability (or instability) may relate to generalization.

6) It appears that the current task design presents the subject with text answers on screen. This is a possible confound, as this visual presentation and the act of reading may impact neural activity. While I do not see this as a major confound, the authors should describe it in the discussion.

7) The authors use the terms "decoding" and "classification" with intended distinct meanings. I suggest that they define these terms very explicitly (i.e. "we will use the term decoding to indicate... vs. classification to indicate") to avoid reader confusion.

8) At the end of the discussion, the authors incorrectly describe a cursor (computer mouse) control based communication prosthesis as a speech prosthesis. This discussion point is misleading, as the "slow" rate mentioned is under very different conditions and directly generalizes to generalized communication, whereas the authors study is highly constrained with respect to the communications possible. This point of comparison is very important, as it places the author's study within the larger field of neurally driven communication prostheses and as such I suggest the authors contextualize their work more accurately:

- In addition to reference 25, the BrainGate group has examples of free-choice, free-pace typing at rates for 6-10 words per minutes (see Nuyujukian, PLoS One and Gilja, Nature Medicine), these systems allow for a much wider range of communications than the system presented in the current study. The PLoS One study also demonstrates other computer tablet usage tasks.

- The current study should be quantified with respect to effective communication rate. For each context, there are a finite possible set of responses, ranging from 2 to 10 or about 1 to 4 bits of information per response. Given the > 30% error rate in the answers, the effective communication bits per response will be lower than this range. The authors should compare the effective communication rate in their system to that of existing communications BCIs.

- These comparisons can also be used to compare other ECoG speech classification papers, including work by the Chang group and Mugler, J Neurosci & JNE).

Reviewer #3:

Remarks to the Author:

In the study by Moses et al. a speech dialogue is decoded from epilepsy patients implanted with ECoG grids in real-time. Speech decoding in humans under these conditions have been done before in real-time and offline. The novel aspect of this study is the online decoding of speech production and perception within one framework. Generally, the study is well written and explains their method in great detail.

Questions & issues:

Decoding of speech perception is interesting scientifically but in a clinical setting why not use a microphone and directly decode the question from that?

In relation to that was there a test on how good the answers could be decoded given the ground truth questions (context)?

Speech production was decoded but for patients who might need a speech prosthesis it would be interesting to decode imagined speech. Have there been attempts to decode imagined speech in this setting or were answers always verbalized?

In the supplementary material why only show participant 2,3 / 1,2 and not all three of them? Is the example one in the main figures the missing one, respectively? If so this could be made clearer.

Some p values for figure 2 are given in the text, why not use * notation in the figure to show significance level?

REVIEWER 1 COMMENTS

(1) The Introduction could be improved by presenting the scientific and technological aspects that need to be investigated to realize the real-time decoding system. This could provide a parallel structure for the respective methods and results.

We have added text to page 2 of the Introduction, which provides a more explicit description about the key technical advances (enabled by a clear scientific understanding) that we believe are necessary for a fully functional real-time decoding system.

Enabled by a clear scientific understanding of neural speech processing, this prosthesis should be capable of performing at least the following five key techniques: (1) acquiring relevant neural signals with sufficiently high spatial and temporal resolution for representing speech; (2) extracting reliable speech-related features from these signals in real-time; (3) detecting when the patient intends to produce speech output; (4) incorporating external cues (such as heard speech) to provide contextually-relevant decoding constraints; and (5) decoding the intended speech output from neural signals.

(2) The presented results in Figures 2 and S3 could be improved by overlaying functional mapping results to better understand the relevance of specific functional areas in the decoding process.

Although we agree that functional mapping results would help illuminate which regions were more or less relevant during speech decoding, we unfortunately do not have functional mapping data available for these participants. We contacted some of the clinicians that worked with these participants, but their notes did not contain any functional mapping information relevant to this work.

(3) The presented results in Figure 3 could be split into the individual steps of the detection and classification process. This could be a supplementary figure that sheds some light on which aspect of this multi-step process requires more training data. It would also make it easier to understand what the accuracies of each of these steps are.

We agree with the Reviewer here that an analysis of how detection performance is affected by the amount of data used to fit the detectors would be interesting. We performed this analysis and found that speech detection performance plateaus at around 25% of the available training data. We include these findings in a new supplementary figure on Page 54 and with revised text on pages 8, 15, and 43.

(4) It would be interesting to understand whether such high spatial electrode density is actually necessary for a speech prosthesis. Perhaps this could be done by averaging across electrodes, which could effectively simulate a less spatially dense grid. The results of this classification simulation could be presented in a figure not unlike Figure 3.

The Reviewer raises an interesting question pertaining to the effect that spatial resolution had on performance. We addressed this by adding a new analysis in which a lower resolution ECoG grid is simulated for one of our participants. Instead of averaging across electrodes, we instead sub-sampled electrodes, which we felt was more similar to what a low-resolution ECoG array would provide (under this assumption, the low-resolution arrays would have similar electrode contact sizes and extent (area) of coverage but with fewer channels). We confirmed that having a lower spatial resolution significantly hinders decoding performance. Additional text was added on pages 9, 15, and 48 to include these findings, and a new figure was added on page 10.

(5) The number of trials for each subjects, and the duration during which they were recorded is not clearly presented in the manuscript.

We have created a new supplementary table summarizing the amount of data and number of trials collected for each participant (on page 58), which is referred to in the text on page 5.

(6) The variability in the utterance could be a limiting factor that warrants a more sophisticated analysis.

The Reviewer brings up an important aspect of speech production that we did not explicitly address in the text of our original manuscript. There is natural variability associated with producing the same utterance multiple times (a single speaker producing the same utterance multiple times will not produce the exact same acoustics each time). This concept is one of the primary motivations for using HMM-based classifiers in this work. During Viterbi decoding, the amount of time each HMM can spend in each phone state is flexible. This results in increased robustness of the classifiers to natural variability in the produced utterances; the stretching and contraction of phones in certain pronunciations should not have a large negative effect on the ability of the classifier to identify the utterance. We have reworded the text on pages 14-15 to make this more clear.

We selected an HMM model architecture for several reasons, with arguably the most important being its inherent robustness to certain kinds of variability in the structure of speech. During Viterbi decoding, the answer classifiers were robust to variability in the exact duration and pronunciations of the produced answers because the amount of time each HMM could spend in each phone state was flexible.

(7) The context integration is necessary for the real-time system. It should be clearly stated that this is only used for the real-time and not for the post-hoc analyses.

We apologize for a lack of clarity regarding where context integration is used in the analyses presented in the paper. The context integration model was used in real-time and in many of the offline analyses, where it was important to evaluate the performance of the full system with optimized decoding parameters. However, it is the case that some of the follow-up analyses did not use the context integration model. We have modified the text on pages 9 and 10 to indicate specifically where context

integration was not used. In addition, we have added a sentence on page 3 indicating that the context integration model was used in all analyses except where explicitly indicated.

This context integration model was used during online real-time decoding and offline analysis (except where specifically indicated).

(8) Section 4.8.1. mentions that “Classification accuracy and cross entropy metrics were computed for each participant by using only the utterance classification and context integration models (and not the speech detection models).” This should be mentioned upfront in the results.

Given multiple comments from the Reviewers, we recognize that it is important for us to make this point clearer. We have now expanded the text on page 6 of the revised manuscript to indicate that we treat “classification” and “decoding” different, with the key distinction being that classification uses true events, while decoding uses detected events. We hope that these changes clarify the issue for readers, as we believe it is an important distinction for evaluating the real-world implications of our approach.

In this work, we explicitly differentiate between the terms “classification” and “decoding”: Given a set of features (such as a time window of neural signals), classification refers to the prediction of a single label from these features, and decoding refers to the prediction of an arbitrary-length label sequence from these features. To evaluate classification performance, we used true event times determined from acoustic transcriptions of the test blocks, ensuring that the appropriate time window of neural signals was associated with each classification target (each test trial).

(9) The Discussion could improve by presenting the necessary scientific and technological steps to translate this demonstration into viable speech prosthesis for locked-in patients. For example, how would the lack of acoustic feedback, and the lack of initial training data affect this approach? What are the remaining scientific and technical hurdles? Which of the many steps in the detection and classification process clearly require improvement.

We thank the Reviewer for this suggestion. Indeed, there are several aspects of speech decoding that may differ substantially between overt and imagined speech. Although there is a small and growing literature on this topic, it is our opinion that many of these questions remain to be addressed and explored systematically. Therefore, we do not know whether phone-based models will be appropriate for imagined speech, or whether it will be feasible to train models on data where we do not know when the patient is imagining speaking. We have expanded the discussion of these topics on pages 16-17 of the revised manuscript.

Ultimately, such a system would also generalize to imagined or covertly-produced speech, particularly for the case of fully paralyzed individuals. There may be additional challenges in translating the approaches presented here to the imagined speech setting; for example, it is unknown whether phone-based models are appropriate for imagined speech, and the specific training procedures used here may need to be modified in cases where patients are unable to speak or move.

REVIEWER 2 COMMENTS

(1) In a few places in the text, the authors use the phrase “high-accuracy rate.” I suggest that the authors dial back the use of the term “high-accuracy” unless they can provide clear justification

and apple-to-apples quantitative comparison to the current state of the art.

Our intent with this wording was to emphasize that the system performs well above chance (indeed, at a level that we feel would be functionally useful to a patient). However, because there has not been (to the best of our knowledge) any previous work describing real-time detection and decoding of produced words and phrases (let alone integration with prior context), we are not aware of an obvious benchmark for us to compare our work to.

To address this concern, we have replaced claims of “high-accuracy performance” with “reliable performance” (see pages 2, 6, 7, and 15). We think that the term “reliable” more accurately represents the state of the decoder, signifying that our system reliably performs above chance.

(2) The authors refer to the task as “naturalistic,” however as described in the methods and in figure 1, the task is externally cued with respect to timing and answers are constrained. While the work conceptually pushes towards naturalistic usage, the study design and the context in which the methods are tested are not naturalistic. This language should be dialed back and the difference between the task and a naturalistic setting should be discussed.

We appreciate the issue the Reviewer has raised here, and we agree that while our paradigm is not entirely natural, it moves the field in the direction of conversational speech, which may be crucial for a functional speech prosthesis. We have edited the text on pages 2 and 13.

(3) There is a high degree of variability in performance across subjects. While I respect the authors statement that it is not possible to make definitive claims about this variability, the paper would be strengthened by elaborating on potential sources of variability in the discussion.

The Reviewer is correct in noting the performance variability across participants and the difficulty associated with pinpointing potential sources of this variability. In addition to the differences in coverage, other factors could cause this variability, including participant alertness and task engagement. We have updated the paragraph on page 16 to include these potential factors.

Other potential sources of variability in decoder performance across the participants include alertness and task engagement.

(4) The authors very responsibly present the detection scores as an aggregate of detection, context integration, and utterance classification. However, it would be helpful to further understand how failures in detection and context (question) identification degrade performance — the authors could compute performance assuming perfect detection and/or context identification. In the case of context identification, performance under hard assignment of context could provide a useful comparison point to the likelihood based soft assignment. This hard assignment could be through the decoder (which would allow us to assess the value of soft assignment) or via an oracle (cheating by using the actual context as presented).

We agree that these kinds of more detailed diagnostics are important for understanding how this decoding system works. We have actually already included the analysis regarding “perfect detection”; it is the classification accuracy analysis, which uses utterance times from the acoustic transcriptions instead of from the speech detector. We have clarified this on page 6 of the revised manuscript and in the Discussion on page 15.

We also agree with the Reviewer that it would be interesting to better understand how variations in context integration affect answer decoding performance. To this end, we performed two new analyses: one in which we performed context integration with “hard” priors (performance under hard assignment of context) and one with “true” priors (the “oracle” approach with the true question being used). The findings were consistent with what we expected: hard assignment did not improve performance, and the true priors only showing improvement in some participants and not in the participant with very accurate question decoding. This is described in the revised manuscript in a new results sub-section on page 13, additional text in the discussion on page 14, added text to the methods on page 27, a description of the statistical tests on pages 48-49, and a new supplementary table on page 61.

(5) Although the supplement is very complete in its description of hyperparameters, the results section describing hyperparameters could be strengthened by providing a concise description of the physiologically relevant hyperparameters (e.g. parameters control time delays, electrode selection). The authors show the importance of hyperparameters, but showing a high degree of variability in performance. Related to this, the authors should provide a description of the stability of tuned hyperparameters as a function of fold — as this stability (or instability) may relate to generalization.

We agree with the Reviewer that our results section could be improved by providing a few examples of the physiologically relevant hyperparameters. We have added some examples to the text in that section on page 8.

We are hesitant to perform an additional analysis to compare hyperparameter values across folds. Because there could be various local optima in the high-dimensional hyperparameter space, it is difficult to directly compare individual hyperparameter values. For example, it is possible that having a long duration neural feature vector with a restrictive p-value threshold performs similarly to a small duration neural feature vector with a lenient p-value threshold (in this scenario, it could be that the classifiers were very sensitive to the total size of the feature vectors). Generally, we think that it is better to compare the *performances* of certain hyperparameter sets during testing rather than directly comparing the individual values in this set. Under this philosophy, a more reasonable approach to measure stability across folds is to assess how well hyperparameters optimized on some test blocks perform on a held-out test block. This is essentially what we have done in the hyperparameter analysis we included here; we showed that the hyperparameter configurations deemed optimal on one set of test blocks also performed well (better than the median for 250 configurations) on held-out blocks. This suggests that the hyperparameters are generalizable to some degree. Although we acknowledge that a more thorough analysis of hyperparameter stability could be performed, we think that this analysis is outside the scope of this project. We primarily wanted to show that improvements that can be gained by performing hyperparameter optimization, hopefully encouraging future work in this field to also perform similar optimizations instead of less quantitative approaches of setting these values (such as trial and error). We have added text to the results section on page 9 to further emphasize that the hyperparameter values were generalizable across test blocks.

Examples of physiologically relevant hyperparameters include a temporal offset shift between perceived and produced phones and the neural data (which could account for neural response delays or speech production planning), the duration of the spatiotemporal neural feature vectors used during model training and testing, and a P-value threshold used when deciding which electrodes should be considered relevant and included in the analyses.

(6) It appears that the current task design presents the subject with text answers on screen. This is a possible confound, as this visual presentation and the act of reading may impact neural activity.

While I do not see this as a major confound, the authors should describe it in the discussion.

The Reviewer is correct to point out that the visual demands of the task, which include reading, could potentially impact the neural activity used to decode perceived and produced speech. We have added text to page 16 mentioning this potential confound.

Additionally, despite using a task that allows participants to choose what to say voluntarily, they were still provided with visual cues listing their response options. While we do not have evidence to suggest that visual presentation or the act of reading influenced the present results, future work could evaluate similar paradigms that do not involve these aspects of visual and language processing.

(7) The authors use the terms “decoding” and “classification” with intended distinct meanings. I suggest that they define these terms very explicitly (i.e. “we will use the term decoding to indicate... vs. classification to indicate”) to avoid reader confusion.

We thank the Reviewer for this great suggestion. It will be helpful for readers to make this distinction more explicit, and we have added text to page 6 to clarify this.

In this work, we explicitly differentiate between the terms “classification” and “decoding”: Given a set of features (such as a time window of neural signals), classification refers to the prediction of a single label from these features, and decoding refers to the prediction of an arbitrary-length label sequence from these features.

(8) At the end of the discussion, the authors incorrectly describe a cursor (computer mouse) control based communication prosthesis as a speech prosthesis. This discussion point is misleading, as the “slow” rate mentioned is under very different conditions and directly generalizes to generalized communication, where as the authors study is highly constrained with respect to the communications possible. This point of comparison is very important, as it places the author’s study within the larger field of neurally driven communication prostheses and as such I suggest the authors contextualize their work more accurately:

- In addition to reference 25, the BrainGate group has examples of free-choice, free-pace typing at rates for 6-10 words per minutes (see Nuyujukian, PLoS One and Gilja, Nature Medicine), these systems allow for a much wider range of communications than the system presented in the current study. The PLoS One study also demonstrates other computer tablet usage tasks.

- The current study should be quantified with respect to effective communication rate. For each context, there are a finite possible set of responses, ranging from 2 to 10 or about 1 to 4 bits of information per response. Given the > 30% error rate in the answers, the effective communication bits per response will be lower than this range. The authors should compare the effective communication rate in their system to that of existing communications BCIs.

- These comparisons can also be used to compare other ECoG speech classification papers, including work by the Chang group and Mugler, J Neurosci & JNE).

We thank the Reviewer for mentioning these crucial points that provide context for our work in the broader literature.

First, we have substantially rewritten the last paragraph of the Discussion to emphasize a distinction between prostheses that allow for communication via non-speech interfaces, and speech prostheses. Our current work is focused on the latter, which has potential advantages for patients in the domain of

naturalness and familiarity (i.e., patients may require substantially less training to learn to use a speech prosthesis compared to a cursor-based prosthesis).

However, naturalness is currently balanced with generalizability, and the Reviewer is correct to point out that other communication prostheses utilize interactions that are currently more amenable to open vocabulary settings. We have noted these points on page 16 of the revised manuscript.

Second, we agree that it is important to provide a more direct comparison among these various neural prosthetic paradigms. Therefore, we have computed information transfer rates (ITRs) for our results (pages 6, 14, 25, 42, and 60). While we agree that ITR is useful for allowing comparison across studies, we also note that it does not fully account for all aspects of the task and analysis. Specifically, we placed a greater emphasis on having a natural task, which created a tradeoff in the size of the vocabulary. We have included a citation that notes that the field could benefit from a revised metric that includes these kinds of qualitative parameters.

Although current state-of-the-art communication prostheses based on letter-by-letter typing, cursor control, and target detection are already beneficial to some patients and are highly generalizable, many are slow and unnatural compared to the type of communication potentially afforded by a speech-based prosthesis, requiring patients to spell out intended messages slowly at rates less than 8 words per minute. An ideal speech prosthesis would be capable of decoding spontaneous, natural speech controlled by a patient's volition and would balance the tradeoff that currently exists in neural prosthetics between generalizability and naturalness.

REVIEWER 3 COMMENTS

(1) Decoding of speech perception is interesting scientifically but in a clinical setting why not use a microphone and directly decode the question from that?

The Reviewer brings up a valid and important point about potentially simpler approaches to decoding heard speech (in this case, the questions). In practice, a microphone could theoretically provide sufficient information for decoding the questions, possible at equal or greater performance than with neural data. However, we believe there are several reasons to consider neural signals for decoding heard speech in a speech prosthesis.

These include the fact that unlike a microphone (or possibly even a microphone with a robust language model attached to the backend automatic speech recognition system), the human brain is unmatched in its ability to not only recognize speech, but interpret it. This is particularly important for ambiguous or noisy speech, which is abundant in natural settings. In addition, neural signals contain information about attention and intention on the part of the listener, which can change the speech they produce in response to what they hear. Without this information, it is possible that context integration models would be less successful.

We have substantially modified the text on page 14 describing these advantages.

In practice, it may be sufficient to use a microphone to recognize heard speech; however, there are several potential advantages to detecting and decoding speech using neural signals. First, speech is often ambiguous; sentences containing the same words in the same order can have completely different meanings depending on the way in which they are spoken. Additionally, speech in natural environments is often masked by irrelevant noise. Both of these issues are challenging for many automatic speech

recognition (ASR) systems. Second, in more complex conversational settings, decoding heard speech from the brain provides important additional information that a microphone signal would not, including whether the listener is attending to the speaker. Finally, there are both scientific and technical advantages to a fully-contained and generalizable speech decoding system that can distinguish among multiple speech sources and use each of them as context for the others. Nevertheless, the context integration approach could be improved by incorporating various input sources as context for large-vocabulary decoding, including video, kinematic, and acoustic sensors placed around the user, and video processing and ASR algorithms could be applied directly to these inputs to improve context decoding.

(2) In relation to that was there a test on how good the answers could be decoded given the ground truth questions (context)?

This is a great point that was also brought up by one of the other Reviewers. Please see our response for Reviewer 2, Question (4).

(3) Speech production was decoded but for patients who might need a speech prosthesis it would be interesting to decode imagined speech. Have there been attempts to decode imagined speech in this setting or were answers always verbalized?

This is an important consideration for many potential applications of a speech prosthesis. While there has been some interesting work on this topic in the literature, we have not yet applied our methods to the imagined speech decoding problem. However, to ensure that our work is contextualized properly, we have added text to pages 16-17 of the Discussion mentioning imagined speech as an ultimate goal. Please see our response for Reviewer 1, Question (9).

(4) In the supplementary material why only show participant 2,3 / 1,2 and not all three of them? Is the example one in the main figures the missing one, respectively? If so this could be made clearer.

We apologize for this confusion. We have updated both the main and supplementary figure captions to note that the participants not shown in the supplementary figures are shown in the main figures (pages 8, 9, 52, and 53).

(5) Some p values for figure 2 are given in the text, why not use * notation in the figure to show significance level?

We are not 100% confident that we have interpreted this comment correctly, but we think that the Reviewer is asking why we list the exact p-values for some of the results are given in the text (and are also given in a supplementary table). We think that for increased transparency, reporting exact p-values in the text alongside the statistical effects found is preferred. We still do include the * notation in the figures where appropriate. Additionally, the publishing guidelines for Nature Communications submissions request that exact p-values are included in the manuscript where appropriate.

Reviewers' Comments:

Reviewer #1:

Remarks to the Author:

The authors have addressed issues 1-9 and have provided additional materials to better present their methods and results. On reviewing these additional materials, I have identified two more major issues that needs to addressed:

Major Issues:

(10) The extremely low discriminative power for the "Speech Event" detection (< 0.01) in Figure S3 raises concerns about the ability to detect the speech event from the ECoG signals in single trials.

(11) The effect of the amount of training data on the speech detection in Figure S5 suggests that "0%" of the data is needed to train the detector for subject 2. Together with the extremely low discriminative power presented in Figure S3 this seems unlikely.

Reviewer #2:

Remarks to the Author:

The authors provided a detailed and satisfactory response to all of my concerns. I appreciate the effort and care taken with the revision. The manuscript, in my opinion, is clear, original, and impactful.

Reviewer #3:

Remarks to the Author:

My points from the last review round have been adequately addressed. I have no further comments.

REVIEWER 1 COMMENTS

(10) The extremely low discriminative power for the “Speech Event” detection (< 0.01) in Figure S3 raises concerns about the ability to detect the speech event from the ECoG signals in single trials.

This comment and the following comment seem to stem from a misinterpretation of how we evaluate the speech detection model.

In the paper, we present two types of analyses that characterize the performance of the speech detection model: (1) Detection Scores, which quantify the accuracy of the model across all timepoints, and (2) discriminative power, which provides a visualization of the relative contributions of individual electrodes to the event detection procedure. It is important to stress that the Detection Scores are the only metric we present to quantify the performance of these models. In contrast, the values for discriminative power are essentially arbitrary, and bear no relationship to the actual performance of the models.

To provide further clarification, we have now added text to page 7 of the revised manuscript that describes how discriminative power for all of our likelihood models is estimated using a calculation intended to resemble a coefficient of determination (r-squared) calculation. We used this estimate because, to the best of our knowledge, there is no standard way of measuring electrode weights for a PCA-LDA model. The discriminative power values do not mean much in isolation; they are primarily useful for quantifying and comparing the **relative** contributions of the electrodes. These discriminative powers were used to visualize where the important electrodes were spatially located for each participant. Because the magnitudes of these values are difficult to interpret in isolation, the fact that the values for the speech detection models were below 0.01 does not provide any useful information about the performance of the speech detection model. Please see pages 26-27 for more details about this computation.

In contrast, all of the detection scores were computed by considering the individual trials during the test blocks. In figures 2 and S3, we show these detection scores. These scores are as high as they are specifically because our speech detector can detect speech from ECoG signals during single trials. The detection score metric not only considers how well the detector can detect speech events broadly throughout an entire test block, it also considers how well the detector can identify individual frames in the ECoG activity that occur during speech perception or production. Please see pages 25-26 for more details about how the detection scores are computed.

Although the absolute magnitudes of these values are difficult to interpret in isolation, the spatial distribution of discriminative powers across electrodes indicates which brain areas were most useful for decoding.

(11) The effect of the amount of training data on the speech detection in Figure S5 suggests that “0%” of the data is needed to train the detector for subject 2. Together with the extremely low discriminative power presented in Figure S3 this seems unlikely.

We apologize for the potential confusion here. The left-most dot in each of these figures was computed using 1% of the available data, not 0%. All of the data points, including this 1%, were computed using stratified samples of the training data to ensure that there were enough samples of speech production, speech perception, and silence. The full details of the procedure we used to compute this figure are given on page 43. As mentioned in the previous comment, the discriminative powers presented in Figure S3 should not be used to assess the performance of any of our models.

We have modified the text in the caption of Figure S5 to provide further clarification about which percentages were used during the associated analyses.

Each plot shows question and answer detection scores (mean with standard error) after fitting the speech detection models with various percents of the available speech and silence data points (between 1% and 100%).